# Cathepsin B-Cleavable Cyclopeptidic Chemotherapeutic Prodrugs

**DOI:** 10.3390/molecules25184285

**Published:** 2020-09-18

**Authors:** Viktorija Herceg, Jordan Bouilloux, Karolina Janikowska, Eric Allémann, Norbert Lange

**Affiliations:** Laboratory of Pharmaceutical Technology, School of Pharmaceutical Sciences, ISPSO, University of Geneva, Rue Michel-Servet 1, CH-1211 Geneve, Switzerland; herceg.viktorija@yahoo.com (V.H.); jordan.bouillouxmaluret@gmail.com (J.B.); karolina.olga.janikowska@gmail.com (K.J.); eric.allemann@unige.ch (E.A.)

**Keywords:** RAFT, cathepsin B, prodrug, doxorubicin, poly (ethylene glycol)

## Abstract

Cyclopeptidic chemotherapeutic prodrugs (cPCPs) are macromolecular protease-sensitive doxorubicin (DOX) prodrugs synthesized from a cyclodecapeptidic scaffold, termed Regioselectively Addressable Functionalized Template (RAFT). In order to increase the chemotherapeutic potential of DOX and limit its toxicity, we used a Cathepsin B (Cat B)-sensitive prodrug concept for its targeted release since this enzyme is frequently overexpressed in cancer cells. Copper-free “click” chemistry was used to synthesize cPCPs containing up to four DOX moieties tethered to the upper face of the scaffold through a Cat B-cleavable peptidic linker (GAGRRAAG). On the lower part, PEG 5, 10 and 20 kDa and a fifth peptidyl DOX moiety were grafted in order to improve the solubility, bioavailability and pharmacokinetic profiles of the compound. In vitro results on HT1080 human fibrosarcoma cells showed that cPCPs display a delayed action that consists of a cell cycle arrest in the G2 phase comparable to DOX alone, and increased cell membrane permeability.

## 1. Introduction

Chemotherapy remains the first option for cancer treatment in most patients. Its goal is to deliver a maximal dose of an anti-cancer agent to the tumor site with minimal toxicity to other tissues. Practically, most chemotherapeutic drugs are of low molecular weight and low water solubility. Once administered, they do not differentiate between tumor and healthy cells, leading to severe side effects. To overcome these drawbacks, macromolecular (mostly polymeric) prodrug systems composed of a scaffold to which an anti-cancer drug is attached were explored [1]. Macromolecular prodrug systems aim at increasing the solubility of poorly water-soluble drugs, improving their pharmacokinetics, optimizing their half-life in the blood and finely tuning their release at the target site [1,2,3]. However, one of the main shortcomings that prevents their implementation into clinical practice comes from the polydispersity of polymers employed in their preparation and the inability to precisely regulate the number and position of the attached biologically active moieties. In order to tackle this problem, we selected a Regioselectively Addressable Functionalized Template (RAFTs, see Figure 1). These RAFTs are macromolecular scaffolds containing two separate addressable domains. The spatial separation of these domains, the upper and the lower face, enables tailored attachment of different functionalities [4]. The RAFT concept has shown to be a versatile tool for various applications. Since RAFT enables the spatial separation and arrangement of protein secondary structures, RAFT templates were primarily developed—and are mostly used—in peptide mimicry and molecular recognition studies [5,6,7,8]. However, in the last decade, RAFT macromolecular constructs found their function in a wide variety of biological applications. For instance, tetramer RGD-containing RAFTs have shown in vivo efficacy in the targeting of α_υ_β_3_ integrins [9] and delivering imaging agents and drugs [10,11], and RAFTs coupled with biotin, fluorescein [12], Tc^99m^ [13], Cy5 [14] or positron emission tomography agents [15] were evaluated for the molecular imaging of cancer. Furthermore, RAFT templates have been used for the development of anti-cancer vaccines [16], sensors and as redox-active materials [17].

In our approach, we used the RAFT scaffold for the development of protease-sensitive cyclopeptidic chemotherapeutic prodrugs (cPCPs) aimed at providing targeted delivery of doxorubicin (DOX), a potent anti-cancer drug. DOX (Figure 1) is an anthracycline antibiotic used in the treatment of a large number of pathologies including acute myeloblastic leukemia, acute lymphocytic leukemia, bladder cancer, breast cancer, gastric cancer, head and neck cancer, squamous cell carcinoma, Hodgkin’s and non-Hodgkin’s lymphoma, both small and non-small lung cancer, neuroblastoma, testicular cancer, thyroid cancer, Wilms’ tumor and different types of sarcomas [18,19]. Its action consists of the intercalation into the double-stranded DNA helix [20], inhibition of topoisomerases I and II [21,22], metal-ion chelation [23] and creation of free radicals [19,24]. Overall, DOX acts by provoking cell death by multiple mechanisms that may vary between cell types from apoptosis to necrosis [25]. Moreover, the presence of fluorescent hydroxyl-substituted anthraquinonoid chromophore [26] makes DOX a perfect candidate for the development of theranostic agents. Although being a drug of choice for the therapy of a large number of diseases, DOX treatment may provoke cell resistance [27] and severe side effects, from which cardiotoxicity accounts for up to 30% of death in cancer patients receiving this drug [28].

In order to prevent the side effects, in this article, we exploited the protease-sensitive prodrug approach. It makes use of the increased and aberrant proteolytic activity in certain cells and tissues to enable the active targeting and the controlled release of a drug inside diseased tissue sparing the healthy one. Cathepsin B (Cat B) has already demonstrated to be an efficient target in DOX delivery [29]. Cat B is a lysosomal cysteine protease whose upregulated activity is associated with the progression of breast cancer, ovarian, colorectal, lung, pancreatic adenocarcinoma and others [30,31,32] Cat B secreted by exocytosis may facilitate cancer progression and metastasis development by degradation of the extracellular matrix (ECM) and by activation, processing and degradation of growth factors, cytokines and chemokines. Furthermore, in some cases, Cat B activity may serve as a prognostic indicator of disease recurrence and overall patient survival [30,31,32].

In this work, we present the synthesis, fluorescence properties, cleavage and in vitro evaluation of cPCPs with up to five DOX moieties attached to the upper side of the RAFT template via the Cat B-cleavable peptide linker (GAGRRAAG), and one or two high molecular poly (ethylene glycol) (PEG) side chains attached on the lower part (Figure 1).

## 2. Results and Discussion

### 2.1. Syntheses

RAFT scaffolds containing DOX bound via the Cat B-cleavable spacer were prepared by a five-step synthetic procedure employing copper-free click chemistry, namely strain-promoted alkyne-azide cycloaddition. The initially protected Lys residues and click chemistry approach allowed the synthesis of conjugates with a precisely defined composition. Firstly, the N-terminal of the Cat B-cleavable peptide was functionalized with DBCO-NHS ester and its carboxylic end was attached to the amino group of DOX by standard peptide chemistry. An example of the generic synthesis pathway for tetrasubstituted-diPEGylated conjugates is presented in Scheme 1 (for other generic synthesis pathways, see Appendix A). ESI-MS and NMR analyses confirmed the identity of the compound, *m*/*z* 1528.85 [M + H]^+^, which corresponds well with the calculated value of 1528.65. In the case of PEG-containing compounds, unprotected Lys residues on the RAFT scaffolds were functionalized with DBCO-NHS ester and PEG polymers of different molecular weight (5, 10 or 20 kDa) were attached. Depending on the type of RAFT and the number of unprotected Lys, two PEG 5 kDa, two PEG 10 kDa or one PEG 20 kDa were attached. Next, Boc-protected Lys were deprotected and functionalized with APA. The azido-modified Lys residues were then “clicked” to the dibenzylcyclooctyne-functionalized Cat B linker bound to DOX. MALDI-TOF analysis after the semi-preparative RP-HPLC purifications confirmed the identity of the final compounds: 17395.7 (*m*/*z*) for cPCP_4/5_^2^, 28394.9 (*m*/*z*) for cPCP_4/10_^2^, 24219.8 (*m*/*z*) for cPCP_2/20_ and 9073.4 (*m*/*z*) for cPCP_5_ (Table 1).

Table 1 summarizes the properties of the four obtained RAFT-based conjugates, with their name, molecular weight and number of DOX and PEG moieties attached. Structures can be seen in Scheme 1, Appendix A. ESI-MS, NMR and MALDI-TOF data can be found in the Appendix A).

To study the effect of PEG on the biological properties of DOX bound to the Cat B-cleavable peptide linker, RAFTs functionalized with five DOX bound via the Cat B-sensitive linker and PEG_10_-GAGRRAAG-DOX were also synthesized.

### 2.2. Fluorescence Quenching

The quenching effect of closely packed DOX molecules in RAFT conjugates was assessed by measuring the decrease in the fluorescence intensity compared to free DOX in solutions. As it can be seen in Figure 2, with the exception of cPCP_4/5_^2^, fluorescence intensities decrease significantly with increasing numbers of DOX molecules coupled to RAFT or with the increase in the molecular weight of PEG used in the construct. The highest fluorescence quenching, a 47.1 ± 8.4-fold decrease in fluorescence intensity, was observed in cPCP_5_ which contains the highest number of DOX molecules, followed by 44.0 ± 11.7 seen in cPCP_4/10_^2^ and 28.3 ± 1.6 in cPCP_2/20_ both with high-molecular weight PEG attached. The lowest quenching was measured in cPCP_4/5_^2^, at 6.1 ± 0.4.

### 2.3. Cleavability with Cat B from Human Placenta

The time-dependent increase in fluorescence emitted by DOX after prodrug activation by Cat B from human placenta is depicted in Figure 3. It can be noted that all four DOX-containing RAFT conjugates, cPCP_4/5_^2^, cPCP_4/10_^2^, cPCP_2/20_ and cPCP_5_, were efficiently cleaved by Cat B from human placenta at pH 7. In the case of both cPCP_4/5_^2^ and cPCP_2/20_, DOX fluorescence intensity increased up to 120 min arriving at a plateau. However, the highest fluorescence increase, 12.9 ± 3.8-fold that saturated at 180 min, was observed for cPCP_4/10_^2^. The enzymatic activation of the conjugate with the highest number of DOX and the highest quenching factor was not terminated after 240 min. Control samples, incubated without a Cat B presence, did not show DOX release over time (Figure 3 inlet).

### 2.4. Cytotoxicity of cPCPs

The inhibition of proliferation of cPCPs containing DOX bound via a Cat B-cleavable peptide linker was assessed in HT1080 human fibrosarcoma cells after 72 h of incubation by the WST-1 assay based on the quantification of formazan produced in metabolically active cells.

Concentrations up to 50 μM (DOX eq) of cPCP_4/5_^2^, cPCP_4/10_^2^ and cPCP_2/20_ were tested. IC_50_ values for cPCP_4/5_^2^, cPCP_4/10_^2^ or cPCP_2/20_ were not assessed since cytotoxicity (around 50%) was only observed after 72 h in cells incubated with 50 μM (DOX eq). At this concentration, no significant difference in cytoxicity between the three PEG-containing cPCP conjugates could be observed (Figure 4, upper right corner). In order to evaluate if PEG could cause the low cytoxicity of DOX-containing cPCPs, three additional compounds were tested, cPCP_5_, PEG_10_-GAGRRAAG-DOX and DBCO-GAGRRAAG-DOX. While no cytotoxicity could be observed after 72 h in cells incubated with PEG_10_-GAGRRAAG-DOX (Appendix A), DBCO-GAGRRAAG-DOX (Appendix A) displayed significant cytotoxicity. The results of the assay with cPCP_5_ are presented in Figure 4. The calculated IC_50_ value for cPCP_5_ was 0.86 μM (DOX eq). To compare, the IC_50_ of free DOX was found to be 0.16 μM after 72 h of incubation (data not shown).

### 2.5. Uptake and Intracellular Accumulation of DOX-Containing cPCPs

Confocal fluorescence micrographs were taken at 1, 3, 24, 48 and 72 h upon exposure to 0.3 μM of free DOX (Figure S data not shown) or 30 μM (DOX eq) of cPCP_4/5_.

In order to better understand the cellular fate of DOX-containing cPCPs, lysosomes were labeled with LysoTracker^®^ Green and the nuclei were stained with Hoechst 33342. Fluorescence micrographs showed the enhanced number of lysosomes (green) and intense red fluorescence in the cytoplasm of cells incubated with cPCP_4/5_^2^ at all time points (Figure 5).

PCC was calculated between the DOX-emitted fluorescence and LysoTracker^®^ Green at all time points. After 1 h of incubation with cPCP_4/5_^2^, PCC was 0.77 ± 0.06, indicating the uptake of the compound and its localization in the late endosomes and lysosomes. At 3 h, PCC slightly increased to 0.80 ± 0.06. Calculated PCC was 0.91 ± 0.02 at 24 h, 0.88 ± 0.01 at 48 h and 0.89 ± 0.03 at 72 h, still localizing DOX in the lysosomes. Moreover, at 24 h of incubation with cPCP_4/5_^2^ and free DOX, PCC was calculated between the Hoechst 33342 and DOX to put into context the DOX-emitted fluorescence with the DNA. However, no correlation was found between DOX and Hoechst 33342 at 24 h in the cells incubated with cPCP_4/5_^2^, indicating that there was no DOX in the nuclei despite the high concentration of DOX in the cells incubated with the cPCP_4/5_^2^ conjugate (Appendix A). This result was confirmed after 48 and 72 h of incubation when no red fluorescence was observed in nuclei of the cPCP_4/5_^2^-treated cells. To compare, at 24 h, PCC between DOX and Hoechst 33342 in the cells incubated with free DOX at 0.3 μM concentration was 0.41 ± 0.10 and cell nuclei were visibly red in the analyzed micrographs (Appendix A).

### 2.6. Differences between the Intracellular and Nuclear DOX Fluorescence Measurements

In order to investigate the release of DOX from cPCPs, DOX internalization into HT1080 cells and into the nucleus was monitored using flow cytometry. First, total intracellular DOX fluorescence, which comprises fluorescence found in the cell cytoplasm, membranes and nucleus, was measured. For that purpose, cells were incubated for 24 h with 0.3 μM of free DOX or equimolar concentration (DOX eq) of cPCP_4/5_^2^, cPCP_4/10_^2^, cPCP_2/20_ or cPCP_5_. To identify the long-term efficacy of DOX release from PEG-containing cPCPs and cPCP_5_ without PEG, cells were treated with cPCP_4/5_^2^ or cPCP_5_ for 48 and 72 h. After 24 h of incubation, cellular accumulation of DOX was observed with all tested compounds. However, although an equimolar concentration (DOX eq) was used, DOX-emitted fluorescence was significantly lower in all cPCPs compared to the DOX control group (Figure 6A). While the increase in intracellular DOX fluorescence at 24 h in cells treated with cPCP_4/5_^2^, cPCP_4/10_^2^, cPCP_2/20_ and cPCP_5_ suggests the internalization of the conjugates, only cells treated with cPCP_5_ showed a significant increase in DOX accumulation in the nucleus (Figure 6B). Results obtained after 48 and 72 h of incubation for cPCP_4/5_^2^ display the same trend (Figure 6C). We can note that DOX-emitted fluorescence is observed globally inside cells, but very little is found specifically in the isolated nuclei. In contrast, data acquired with cPCP_5_ conjugate show the accumulation of DOX both in the cell cytoplasm and in the nuclei (Figure 6D).

### 2.7. Cell Cycle Analysis

Hoechst 33342, a fluorescent dye that intercalates between A and T bases, was used to monitor the changes in DNA content and distinguish between different phases of the cell cycle. Results obtained after 24 h of incubation of HT1080 cells with either free DOX or equimolar concentration (DOX eq) of cPCP_4/5_^2^, cPCP_4/10_^2^, cPCP_2/20_ or cPCP_5_ are presented in Table 2. In order to follow the changes in cell cycle distribution over a longer period of time, additional analysis was done with cells incubated for 48 and 72 h with free DOX, cPCP_4/5_^2^ or cPCP_5_. As it can be noted from Table 2, DOX treatment provokes cell cycle arrest at the G2 phase in HT1080 cancer cells. After 24 h of incubation with free DOX, there is a reduction in the number of cells in the G1 phase of 30.4 ± 2.3% and an increase in S of 36.7 ± 11.9 and G2 of 31.1 ± 14.8% compared to 51.5 ± 3.7% in G1, 25.6 ± 3.0 in S and 21.0 ± 0.4% in G2 found in the control. Although the reduced Hoechst fluorescence intensity indicates the accumulation of DOX in the nuclei, no significant differences were observed at this time point with PEG-free cPCP_5_. This suggests the delayed action of cPCP_5_ in comparison to free DOX. Results obtained at 48 and 72 h show an alteration in the cell cycle for cells incubated with cPCP_5_, a significant reduction in the number of cells in the G1 phase and an accumulation of cells in the G2 phase, comparable to free DOX (Appendix A). No alteration of the cell cycle is detected in cells incubated with any of the PEG-containing cPCPs that correlates with the results of DOX uptake into the nucleus.

### 2.8. Discussion

In this article, we present, for the first time, defined protease-sensitive prodrugs for the targeted delivery of anti-cancer agents. Cyclopeptide RAFTs are suitable to act as a drug delivery scaffolds because of their synthetic flexibility and easy control of the size and spatial orientation of biologically active moieties by incorporation of amino acids with orthogonally protected side chains. In order to obtain macromolecules with a similar shape and steric hindrance that would enable the loading of several anti-cancer drug moieties with an altered pharmacokinetic profile, hence diminishing the systemic toxicity of DOX, we synthesized a RAFT containing two to five DOX molecules coupled via a Cat B-cleavable peptide linker (GAGRRAAG) (Table 1). The application of RAFT scaffolds enabled the controlled stepwise ligation of two different functional groups, PEG and DOX-Cat B-sensitive conjugate, on the two opposite faces of the cyclopeptidic plane. Copper-free click chemistry by strain-promoted alkyne-azide cycloaddition proved to be an excellent choice because of its low toxicity, compatibility with in vitro assays and relatively high reaction yields [33]. Although schematically the upside-down strategy might be more convenient for the attachment of high-molecular weight elements [17], we chose the schematically downside-up approach by first attaching PEG to the down-facing side of the scaffold and then the DOX-peptide conjugate on the upper face. Though PEG conjugation to the RAFT was successful, reactions of DBCO-GAGRRAAG-DOX with RAFT containing two PEG 5 kDa or 10 kDa led to the formation of side products containing two, three or four DOX conjugates per cPCP. These were successfully separated using RP-HPLC and identified by MALDI-TOF. The reaction incompleteness might have been caused by the steric hindrance of the high-molecular weight PEG hampering the reaction between DBCO-GAGRRAAG-DOX and the azide moieties coupled to Lys residues.

DOX was chosen because of its potency as an anti-cancerous agent and its fluorescent properties that can serve as a reporter of the released molecules for theranostic purposes. As already described in the literature, DOX fluorescence intensity depends on several factors: DOX concentration, microenvironment and, in in vitro conditions, upon DOX interaction with various cellular components [26,34]. The fluorescence quenching effect of closely packed DOX molecules and its correlation with high-molecular weight PEG was assessed and is shown in Figure 2. DOX fluorescence in cPCPs conjugates was quenched in the following order cPCP_5_ > cPCP_4/10_^2^ > cPCP_2/20_ > cPCP_4/5_^2^. As expected from our previous results [35,36], the highest loading with DOX units also resulted in the highest quenching ratio as observed for the conjugate cPCP_5_. The data indicate that in cPCP conjugates, quenching of DOX molecules increases with the number of coupled DOX and the molecular weight of PEG used. It can be noted that in the conjugates cPCP_4/10_^2^ and cPCP_4/5_^2^ that contain the same number of DOX molecules, longer PEG chains “squeezed” DOX together and the quenching effect was more pronounced. We have already shown that an important number of different RAFT conjugates can be designed with up to six peptide motifs per carrier [35,36]. It is expected that higher loading with DOX will also result in higher quenching. Furthermore, the molecular weight of the modulating PEG moieties can easily be changed and its effect could be exploited in future studies.

Our goal was to use the elevated levels and deregulated Cat B activity in cancerous tissues as a mediator for targeted drug release [29]. As it was observed in numerous studies, the rate of DOX release is governed by the length and the structure of the peptide linker, and its steric interaction with Cat B [26,37] that, in small molecule substrates, preferentially cleaves R-R bonds from the Arg’s carboxyl-end [38]. The GAGRRAAG peptide sequence was designed from the AGRRAA sequence identified as a Cat B-specific substrate by Ruzza et al. [39]. Two G residues (at the amino- and carboxyl-end) were inserted for further chemical modification. Proteolytic cleavage of the GAGRRAAG peptide linker and the release of DOX from all cPCPs were demonstrated with Cat B from human placenta (Figure 3).

The cytotoxic potential of DOX-containing RAFT conjugates on HT1080 cells was assessed using a WST-1 cell proliferation assay. In order to show the feasibility of our approach, we have chosen this cell line for its relatively abundant expression of the targeted enzyme with respect to other human cancer cells [40]. In order to further validate our methodology, a comparison to other cell lines with different protease expressions should be evaluated. Results showed that all PEG-containing RAFT conjugates were very well tolerated by cells and exhibited very similar cytotoxicity profiles over the 72-h incubation period. At the highest tested concentration of 50 μM (DOX eq), which is over 300-fold higher than the IC_50_ observed with free DOX, cell viability remained around 50%. The assumption that PEG might interfere with DOX cytotoxicity was confirmed by the assessment of the cytotoxicity of cPCP_5_ and PEG_10_-GAGRRAAG-DOX. Meanwhile, the IC_50_ at 72 h of cPCP_5_ that does not contain PEG was 0.86 μM (DOX eq), which is only 5.4 times higher than the IC_50_ observed with free DOX. The interference of PEG with potential cytotoxicity was also confirmed with PEG_10_-GAGRRAAG-DOX and DBCO-GAGRRAAG-DOX. These results are in agreement with reports on other described Cat B-cleavable DOX macromolecular prodrugs [37,41,42] in which the IC_50_ was well above the value observed with free DOX. In these studies, however, in vitro cell viability did not necessarily correlate with the in vivo anti-tumor efficacy [43].

Since no correlation of DOX release studies and the cytotoxicity results was found, alternative techniques were investigated to get to an understanding of the in vitro results. Confocal fluorescence micrographs of cells incubated with cPCP_4/5_^2^ and labeled with LysoTracker^®^ Green showed the fast internalization and strong colocalization between DOX found in cPCP_4/5_^2^ and lysosomes. Calculated PCC, between DOX and LysoTracker^®^ Green, was 0.77 ± 0.06 after one hour and reached 0.80 ± 0.06 after 3 h of incubation. Although being a lysosomal protease, Cat B is known to be excreted by HT1080 cells [40]. Since media were replaced at the time of treatment with cPCP_4/5_^2^, we believe that very low amounts of DOX were released extracellularly. At 24, 48 and 72 h of incubation (Figure 5), calculated PCC was 0.91 ± 0.02 and it remained at this level up to 72 h. These findings may account for the low cytotoxicity of the pegylated compounds since the loss of the therapeutic effect can be associated with the stable entrapment of the conjugates in the lysosome [44]. This has been supported by the lack of visible DOX-emitted fluorescence in cell nuclei after 24, 48 and 72 h of incubation. To compare, at 24 h, nuclei of cells incubated with the 0.3 μM concentration of free DOX were visibly red (Appendix A). The relatively low PCC of 0.41 ± 0.10 calculated between DOX and Hoechst 33342 at that time point in cells incubated with free DOX is a result of the quenched DOX fluorescence by the molecule’s intercalation into the double-stranded DNA helix [45]. The observed red fluorescence in the nuclei actually came from DOX associated with histones [34].

Free DOX enters into the cell by simple diffusion [19]. Once in the cytoplasm, DOX complexes with the 20S subunit of the proteasome and is internalized into the nucleus by the nuclear pore complex. Due to the higher affinity of DOX towards the DNA than to the proteasome, DOX binds to the DNA [46]. Since DOX fluorescence gets quenched by its intercalation in the DNA [20] and no DOX could be seen in the cell nucleus on confocal micrographs, the intensity of internalization and nuclear uptake was measured by flow cytometry. For easier comparison, cells were incubated with 0.3 μM of free DOX and with equimolar concentrations (DOX eq) of cPCP_4/5_^2^, cPCP_4/10_^2^, cPCP_2/20_ and cPCP_5_. DOX fluorescence intensity is highly dependent on the microenvironment and upon the interaction with various cellular components. Moreover, DOX fluorescence is non-linear and, as it was shown by Mohan et al. [34], it levels off at concentrations higher than 43 μM. With this in mind, cell cytometry measurements of intracellular fluorescence should be taken with precaution since they might not reflect the real DOX concentration. However, taken together, the fluorescence of DOX and Hoechst 33342 measured in the nucleus accounts for the real DOX release. DOX fluorescence intensity in the nucleus depends on mainly two factors, DOX intercalation into the DNA and its interaction with histones [20,34]. Although both the increase in intracellular DOX fluorescence at 24 h in cells treated with cPCP_4/5_^2^, cPCP_4/10_^2^, cPCP_2/20_ and cPCP_5_ and fluorescence micrographs of cPCP_4/5_^2^-treated cells show the internalization of the conjugates, only cells treated with cPCP_5_ and free DOX had significant DOX accumulation in the nucleus (Figure 6B,D). These findings were confirmed by the decrease in Hoechst 33342 fluorescence [45] observed in the cell cycle results of cells incubated with free DOX and cPCP_5_ at all time points. No significant decrease in Hoechst 33342 fluorescence could be seen in the control cells or cells incubated with any of the cPCP conjugates containing PEG (Appendix A).

It is well known that one of the most important mechanisms of DOX action lies in its intercalation into the double-stranded DNA helix and stabilization of a topoisomerase II (TOPO II)-DNA intermediate making double-stranded breaks that fail to be repaired by the cell machinery. This leads to TOPO II-mediated DNA damage and cell cycle arrest [22]. It is important to emphasize that no cell cycle synchronization of HT1080 cells was done in our experiments. Although no alteration was found in cell cycles of cells treated with cPCP_4/5_^2^, cPCP_4/10_^2^ or cPCP_2/20_ at 0.3 μM (DOX eq), a trend can be seen in the number of cells in the different phases of the cell cycle for cells treated with either free DOX or cPCP_5_ at 48 and 72 h which caused cell cycle arrest in the G2 phase.

Although nuclear localization of released DOX was not detected on confocal fluorescence micrographs and no significant DOX accumulation in the nuclei or cell cycle alteration was observed in the low concentration of PEG-containing cPCPs, preliminary flow cytometry results acquired on cells treated with 30 μM (DOX eq) of cPCP_4/5_^2^ showed an increased presence of DOX in the cell nucleus and subsequent cell cycle arrest in the G2 phase (Appendix A). These results and the results obtained with cPCP_5_ lead us to conclude that DOX released from the conjugates displays the same cytotoxicity mechanisms as free DOX, but with a time-delay. Similar results were already observed for the molecular mechanism of cell toxicity for *N*-(2-hydroxypropyl) methacrylamide (HPMA) copolymer-bound DOX via a Cat B-cleavable GFLG linker, where the conjugate initiated cell death by the same mechanism as free DOX, but the appearance of the cell arrest was delayed for 4–6 h [41]. Moreover, we believe that the overall cytotoxicity at high doses of cPCP_4/5_^2^ comes from both DOX intercalation in the double-stranded helix, and the effect on cell membrane permeability by the conjugate which was confirmed by the increased uptake of Draq7, a cell-impermeable dye that only stains the nucleus of permeabilized or dead cells in flow cytometry experiments [47].

The assembly of gathered results led us to conclude that the conjugates containing high-molecular weight PEG are not efficiently cleaved in the in vitro conditions or possibly stay entrapped into lysosomes. We assume that this is due to the excessive steric hindrance caused by PEG.

PEG of various molecular weights has been used in pharmaceutical technology for decades to increase water solubility, reduce immunogenicity and modify pharmacokinetics [48]. With a purpose to identify PEG-DOX conjugates with improved targeting and efficacy, Veronese et al. [37] synthesized a series of monomethoxy-PEG-peptide-DOX conjugates with both linear and branched structures, and with an M_w_ of 5, 10 and 20 kDa. They used various Cat B-cleavable peptide linkers for covalent attachment of DOX to PEG. In their studies, the rate of DOX release from the conjugates depended on the linker used and not on the PEG properties. Interestingly, all of their conjugates were found to be 10–100-fold less toxic than free DOX, which partially correlates with what was seen using cPCPs in the present study. However, in in vivo studies on a B16F10 murine melanoma mouse model, all PEG-DOX conjugates displayed a significantly longer plasma half-life than free DOX and had greater tumor targeting. Finally, PEG_5000_-GFLG-DOX was selected as the most promising candidate because it showed the most favorable tumor/heart concentration ratio. Since it is well known that the in vitro observation might not always reflect what happens in a much more complexed system in vivo, the real potential of our cPCPs conjugates should be re-evaluated by conducting animal studies.

## 3. Materials and Methods

### 3.1. Materials

Cyclopeptidic RAFTs c-(K_Boc_AK_Boc_PGK_Boc_KK_Boc_PG) (cP_1_^4^), c-(K_Boc_KK_Boc_PGK_Boc_KK_Boc_PG) (cP_2_^4^) and c-(K_Boc_AAPGAKK_Boc_PG) (cP_1_^2^), and Cat B-cleavable peptide (GAGRRAAG, TFA salt) were purchased from Caslo Laboratory (Lyngby, Denmark). α-azido-ω-methoxy-PEG (5, 10 and 20 kDa) was obtained from Iris Biotech (Marktredwitz, Germany). Dibenzylcyclooctyne-NHS (DBCO-NHS) and azidoacetic acid-NHS were purchased from Click Chemistry Tools (Scottsdale, AZ, USA). 5-azidopentanoic acid (APA) was obtained from Bachem (Bubendorf, Switzerland). Doxorubicin hydrochloride salt (DOX. HCl) was purchased from LC Laboratories (Woburn, MA, USA). Trifluoroacetic acid (TFA) and N,N′-diisopropylethylamine (DIPEA) were obtained from Acros Organics (Thermo Fisher Scientific, Wohlen, Switzerland). Formic acid (FA, 98–100%) was purchased from Merck (Darmstadt, Germany). *O*-(7-azabenzotriazole-1-yl)-*N*,*N*,*N*′*N*′-tetramethyluronium hexafluorophosphate (HATU) was obtained from GenScript Corporation (Piscataway, New Jersey, NJ, USA). H-Gly-2-ClTrt resin beads (1.1 mmol/g) were purchased from Sigma-Aldrich (Buchs SG, Switzerland). All solvents and other chemicals were purchased from Sigma-Aldrich (Buchs SG, Switzerland) and Acros Organics (Thermo Fisher Scientific, Wohlen, Switzerland) and were used as received.

Cell culture media DMEM (1x) + GlutaMAX™-1 medium (31966-021), Dulbecco’s Phosphate Buffered Saline (DPBS) (14190-094), 0.25% Trypsin-EDTA (1X) (25200-056) and 100 μL/mL streptomycin and 100 IU/mL penicillin (15140-122), Hoechst 33342 stain (62249) and LysoTracker^®^ Green DND-26 (8783) were all purchased from Thermo Fisher Scientific (Zug, Switzerland). Fetal calf serum (FCS) was obtained from Eurobio (CVFSVF00-01, Eurobio, Courtaboeuf, France). Cell Proliferation reagent (WST-1) was purchased from Roche (11644807001, Roche, Sigma-Aldrich, St. Gallen, Switzerland) and MycoAlert™ kit was obtained from Lonza (LT07-418, Lonza, Basel, Switzerland). Cat B from human placenta (C0150) and the surfactant Igepal CA-630 (542334) were obtained from Sigma-Aldrich (Sigma-Aldrich, St. Gallen, Switzerland).

### 3.2. Methods

Semi-preparative reversed-phase high-performance liquid chromatography (RP-HPLC) was performed on an automatic PuriFlash 4100 device with Interchim Software version 5.0x from Interchim (Montluçon, France). Separations were achieved using linear gradients of solvent B (with H_2_O + 0.01% TFA as solvent A and ACN + 0.01% TFA as solvent B) on a Nucleodur^®^ C_18_-Htec column (762556.210; 5 μm, 21 × 250 mm) from Macherey-Nagel (Oensingen, Switzerland) with one of the following systems: System A: 5–100% solvent B in 60 min; System B: 15–100% solvent B in 60 min; System C: 10–100% solvent B in 60 min; System D: 2–90% solvent B in 30 min; and System E: 5–100% solvent B in 45 min. Purification by size exclusion chromatography (SEC) was performed using Sephacryl^TM^ S-100 medium (Amersham Biosciences, Otelfingen, Switzerland). Ultra-high-performance liquid chromatography (UHPLC) analyses were performed on Waters Acquity^TM^ UHPLC from Waters (Milford, Massachusetts, MA, USA) with QuickStart Empower 2 software. Separations were achieved using linear gradients of solvent B (with H_2_O + 0.1% FA as solvent A and ACN + 0.1% FA as solvent B) on a Nucleodur^®^ C_18_ Gravity EC 50/2 column (760079.20; 1.8 μm, 2 × 50 mm) from Macherey-Nagel (Oensingen, Switzerland) with one of the following systems: System a: 5–95% solvent B in 2 min; System b: 10–100% solvent B in 5 min; System c: 30–100% solvent B in 5 min; and System d: 5–100% solvent B in 4 min. DOX fluorescence emission was detected at 480 nm. Freeze-drying was performed using either a Christ Alpha 2-4 LDplus or Christ Alpha 2-4 LSC device (Martin Christ, Germany). Low-resolution mass spectroscopy using electrospray ionization (ESI-MS) was performed on an API 150EX™ instrument (Applied Biosystems/MDS SCIEX). MALDI-TOF analyses were performed on Axima-CFRplus (Shimadzu, Kyoto, Japan) using CHCA as a matrix. NMR analyses were performed on a Bruker 600 MHz AVANCE III spectrometer (Bruker, Fällanden, Switzerland) and the data were processed with MNova 8 software (MestReLab Research, Santiago de Compostela, Spain). Displacements are given as part per million with the peak of the solvent as the internal reference. Chemical structures and reactions were drawn using the ChemDraw version 14.0.0.117 software package.

### 3.3. Synthesis of DBCO-GAGRRAAG

First, GAGRRAAG (TFA salt) (150 mg, 142 μmol), DBCO-NHS (63 mg, 157 μmol) and DIPEA (30 μL, 172 μmol) were stirred in DMF (4 mL) under argon in the dark at room temperature overnight. DMF was then removed under reduced pressure and the crude was purified by RP-HPLC following *System A*. The intermediate product DBCO-GAGRRAAG (**1**) was then lyophilized to give a white solid (79.4 mg, 79.2 μmol, 55.8%). (**1**) was analyzed by UHPLC following *System a* (retention time of 0.7 min) and ESI-MS (*m*/*z* calculated 1002.10, found 1002.75 [M + H]^+^) (see Appendix A). Then, the obtained (**1**) (64.7 mg, 52.6 μmol), DOX (HCl salt) (27.3 mg, 47.0 μmol) and DIPEA (30 μL, 172 μmol) were stirred in DMF (4 mL) for 15 min before adding HATU (17.5 mg, 46.0 μmol). The reaction proceeded under argon in the dark at room temperature overnight. DMF was removed under reduced pressure and the crude was purified by RP-HPLC following System B. The final product (**2**) was then lyophilized, and obtained as a red solid (43.4 mg, 28.4 μmol, 60.5%). (**2**) was analyzed by UHPLC following System b (retention time of 1.34 min) and ESI-MS (*m*/*z* calculated 1528.65, found 1528.85 [M + H]^+^) (see Appendix A and Scheme 2). ^1^ H NMR (600 MHz, DMSO-d_6_) δ 14.06 (s, 1H, CH-C-C-OH), 13.30 (s, 1H, CH_2_-C-C-OH), 7.95–7.85 (m, 2H, CH_3_-O-C-CH-CH-CH), 7.70–7.64 (m, 2H, N-C-CH, CH_3_-O-C-CH), 7.64–7.58 (m, 1H, CH_2_-C-CH), 7.56–7.43 (m, 4H, N-C-CH-CH-CH-CH, OH-CH-CH-NH), 7.41–7.28 (m, 3H, CH_2_-C-CH-CH-CH-CH), 5.48 (s, 1H, CH_2_-C-OH), 5.25 (d, *J* = 3.5 Hz, 1H, O-CH-O), 5.03 (dd, *J* = 14.2, 3.6 Hz, 1H, N-CH_2_), 4.97 (t, *J* = 4.5 Hz, 1H, C-CH-O), 4.86 (t, *J* = 6.0 Hz, 1H, CH_2_-OH), 4.82 (d, *J* = 6.0 Hz, 1H, CH-OH), 4.57 (d, *J* = 6.0 Hz, 2H, CH_2_-OH), 4.33–4.14 (m, 6H, CH-CH_3_, CO-CH-NH), 4.00 (s, 3H, O-CH_3_), 3.73–3.56 (m, 7H, CO-CH_2_-NH, N-CH_2_), 3.39 (d, *J* = 4.4 Hz, 1H, CH-CH-OH), 3.09–3.03 (m, 4H, CH_2_-CH_2_-NH), 3.00 (s, 2H, CH_2_-C-OH), 2.67–2.58 (m, 1H, CH_2_-CO-N), 2.29 (dtd, *J* = 15.4, 7.7, 3.1 Hz, 1H, CO-CH_2_-CH_2_), 2.24–2.12 (m, 2H, C-CH-CH_2_), 2.06 (dq, *J* = 12.9, 5.9 Hz, 1H, CO-CH_2_-CH_2_), 1.85 (td, *J* = 12.9, 3.8 Hz, 1H, CH_2_-CH-NH), 1.82–1.74 (m, 1H, CH_2_-CO-N), 1.70–1.62 (m, 2H, CH-CH_2_-CH_2_), 1.52–1.42 (m, 7H, CH_2_-CH-NH, CH-CH_2_-CH_2_, CH-CH_2_-CH_2_), 1.25–1.17 (m, 9H, NH-CH-CH_3_), 1.14 (d, *J* = 6.4 Hz, 3H, O-CH-CH_3_). All remaining NH protons could not be assigned precisely but are present in the 8.25–7.25 ppm area, mixed with the already assigned aromatic protons (see Appendix A).

### 3.4. General Procedure for the Synthesis of the PEGylated Cyclopeptidic Conjugates

#### 3.4.1. Coupling of the DBCO Anchor(s)

Each considered cyclopeptide, DBCO-NHS (3 equivalents per unprotected lysine group) and DIPEA (6 equivalents per unprotected lysine group) were stirred in DMF (10 to 20 mL) under argon at room temperature overnight. The crude solution was concentrated and purified by RP-HPLC following System C or System D if required. Final products (3) and (10) were then lyophilized to give white solids. Product (3) was analyzed by UHPLC following System c (retention time of 4.55 min) and ESI-MS (*m*/*z* calculated 2052.5, observed 2052.6 [M + H]^+^) (see Appendix A, Scheme 3 and Scheme 4). Product (10) was analyzed by UHPLC following System c (retention time of 2.14 min) and ESI-MS (*m*/*z* calculated 1393.7, found 1394.7 [M + H]^+^).

#### 3.4.2. PEGylation

(3) or (10) and the corresponding α-azido-ω-methoxy-PEG of 5, 10 or 20 kDa (1.2 equivalent per DBCO moiety) were stirred in DMF (1.5 mL) under argon at room temperature overnight. Following the Fmoc peptide synthesis procedure, DBCO-containing H-Gly-2-ClTrt resin beads were used to remove excess PEG after the reaction using a syringe as the container. The remaining liquid then contained only the corresponding PEGylated cyclopeptide. The solvent was removed under reduced pressure, and then products (4), (5) and (11) were solubilized in H_2_O, lyophilized and obtained as white solids. Product (4) was analyzed by UHPLC following System c (retention time of 2.78 min). Product (5) was analyzed by UHPLC following System c (retention time of 2.62 min). Product (11) was analyzed by UHPLC following System c (retention time of 2.49 min).

#### 3.4.3. Deprotection and Coupling of the Azido Anchors

Boc protections of (4), (5) or (11) were cleaved by stirring the conjugate in a 4.0 mL mixture of TFA/DCM (50:50) at room temperature for 30 min. DCM and TFA were evaporated. APA (5 equivalents per deprotected lysine group) was added to HATU (5 equivalents per deprotected lysine group) in DMF (2 mL) for 5 min, then the mixture was added to the corresponding deprotected cyclopeptide in DMF (3 mL). DIPEA (6 equivalents per deprotected lysine group) was added dropwise and the reaction occurred under argon at room temperature overnight. The solvent was removed under reduced pressure, and then the corresponding product was solubilized with the further eluting solvent. This crude was purified by SEC using a Sephadex^®^ G-10 medium, and mQ water as an eluent. Finally, products (6), (7) and (12) were lyophilized and obtained as white solids. Product (6) was analyzed by UHPLC following System c (retention time of 2.64 min). Product (7) was analyzed by UHPLC following System c (retention time of 2.6 min). Product (12) was analyzed by UHPLC following System c (retention time of 2.34 min).

#### 3.4.4. Coupling of the DBCO-GAGRRAAG-DOX Moieties

Products (6), (7) or (12) were then coupled with (2) (2.1 equivalents per azido moiety) in DMF (2 mL) under argon in the dark at room temperature overnight. DMF was removed under reduced pressure, and the crude was purified by RP-HPLC following System C. Final products (8), (9) and (13) were obtained as red solids after lyophilization. Product (8) was analyzed by UHPLC following System b (retention time of 1.41 min) and MALDI-TOF analysis confirmed the mass of obtained conjugate (*m*/*z* 17395.7 [M + H]^+^) (see Appendix A). Product (9) was analyzed by UHPLC following System b (retention time of 1.98 min) and MALDI-TOF analysis confirmed the mass of obtained conjugate (*m*/*z* 28394.79 [M + H]^+^) (see Appendix A). Product (13) was analyzed by UHPLC following System b (retention time of 2.12 min) and MALDI-TOF analysis confirmed the mass of obtained conjugate (*m*/*z* 24219.8 [M + H]^+^) (see Appendix A).

### 3.5. General Procedure for the Synthesis of the Pentasubstituted Cyclopeptidic Conjugate

#### 3.5.1. Deprotection and Coupling of the Azido Anchors

Boc protections of cP_1_^4^ were cleaved by stirring the conjugate in a 2.0 mL mixture of TFA/DCM (50:50) at room temperature for 30 min. DCM and TFA were evaporated. Azidoacetic acid-NHS (2 equivalents per free lysine group) and DIPEA (4.5 equivalents per free lysine group) were added to the deprotected cyclopeptide in DMF (2 mL) and the reaction occurred under argon in the dark at room temperature overnight. The solvent was removed under reduced pressure, and then the corresponding product was purified by RP-HPLC following System C. Final product (14) was obtained as a white solid after lyophilization. Product (14) was analyzed by UHPLC following System d (retention time of 1.47 min) and ESI-MS (*m*/*z* calculated 1435.54, found 1436.95 [M + H]^+^) (see Appendix A and Scheme 5).

#### 3.5.2. Coupling of the DBCO-GAGRRAAG-DOX Moieties

Product (14) was then coupled with (2) (1.65 equivalent per azido moiety) in DMF (2 mL) under argon in the dark at room temperature overnight. DMF was removed under reduced pressure, and the crude was purified by RP-HPLC following System E. Final product (15) was obtained as a red solid after lyophilization. Product (15) was analyzed by UHPLC following System b (retention time of 1.12 min) and MALDI-TOF analysis confirmed the mass of obtained conjugate (*m*/*z* calculated 9073.61, found 9073.4 [M + H]^+^) (see Appendix A).

### 3.6. Fluorescence Quenching

The fluorescence intensities of free DOX (DOX.HCL) and of equimolar solutions (DOX eq) of cPCP_4/5_^2^, cPCP_4/10_^2^, cPCP_2/20_ and cPCP_5_, prepared in DPBS containing 1% of dimethylsulfoxide (DMSO), were measured at 37 °C using a Safire plate reader (Tecan, Männedorf, Switzerland). Excitation wavelength was set to 480 ± 5 nm and emission to 590 ± 5 nm. The fluorescence quenching factor, defined as an X-fold decrease in background subtracted fluorescence intensities at 590 nm, was calculated with respect to free DOX. All measurements were performed in triplicates and the experiment was repeated three times.

### 3.7. Evaluation of DOX Release by Cat B

Enzymatic activation of cPCPs prodrugs was tested as follows: compounds cPCP_4/5_^2^, cPCP_4/10_^2^, cPCP_2/20_ and cPCP_5_ were solubilized in DPBS (pH 7) containing 1% DMSO and placed in a 96-well plate (clear bottom black plate, 3603, Corning, Luzern, Switzerland). Then, 0.15 U of Cat B solubilized in mQ water was added into each well to a final concentration of 10 μM (DOX eq). Fluorescence intensity measurements were performed at 5, 10, 15, 30, 60, 120, 180 and 240 min upon incubation with Cat B using the Safire plate reader. During the experiment, all samples were kept in the dark at 37 °C. Excitation wavelength of DOX was set to 480 ± 5 nm and emission to 590 ± 5 nm. All the measurements were performed in triplicates and the increase in fluorescence was calculated from the formula below.
$$F=\frac{F_{T}-F_{0}}{F_{0}}$$
where *F* is the X-fold fluorescence increase, $F_{T}$ is the fluorescence of the sample emitted at a certain time and $F_{0}$ is the fluorescence before enzymatic digestion.

### 3.8. In Vitro Cell Studies

#### 3.8.1. Cell Culture

Human fibrosarcoma cell line HT1080 (ATCC^®^ CCL-121™, Manassas, VA, USA) was grown as monolayers in DMEM (1x) + GlutaMAX™-1 medium supplemented with 10% fetal calf serum and 100 μL/mL streptomycin and 100 IU/mL penicillin. Cells were cultivated at 37 °C in humidified 95% air and 5% CO_2_ atmosphere and routinely maintained by serial passage in a new medium. Prior to experiments, cells were tested for mycoplasma contamination with the MycoAlert™ kit.

#### 3.8.2. Cytotoxicity Assays

Cells were seeded at a density of 8000 cells/well in a 96-well plate (clear-bottom black plate, 3603, Corning, Luzern, Switzerland) one day prior to the experiment. For experiments, the complete medium was replaced with serum-free medium containing increasing concentrations of either DOX or DOX containing cPCPs. Cell proliferation assays were performed at 72 h upon incubation using a Cell Proliferation reagent (WST-1) following the manufacturer’s recommendations. The absorption was measured at 450 nm with a plate reader (Safire). The percentage of cell survival was calculated with respect to control samples treated with either medium or a solution of DMSO (50%) in serum-free medium, as follows: [A (test-conc.) − A (100% dead)/A (100% viable) − A (100% dead)] × 100. Mean values from three wells were determined and expressed as ± S.D. All assays were done in triplicates.

#### 3.8.3. Confocal Fluorescence Microscopy of cPCP Uptake and Accumulation in Cells

HT1080 cells were seeded one day before the experiment at a density of 78,500 cells in 1.3 mL of complete medium into a glass-bottom microwell dish (35 mm petri dish, 14 mm microwell, MatTek Corporation). In order to investigate the subcellular localization of DOX fluorescence, cell monolayers were incubated for 1, 3, 24, 48 and 72 h in the dark at 37 °C with 0.3 μM of DOX or 30 μM (DOX eq) of cPCP_4/5_^2^ in a serum-free medium. Control cells were incubated in serum-free medium under the same experimental conditions. Cells were then washed with DPBS and incubated for 20 min with 1 μg/mL Hoechst 33,342 stain and 70 nM of LysoTracker^®^ Green DND-26. Prior to imaging, cells were washed with DPBS and serum-free medium was put into every dish. Fluorescence imaging was carried our using a Zeiss LSM780 (Zeiss, Jena, Germany) inverted confocal microscope. The microscopic images were using 405 and 488 nm lasers as excitation wavelengths, 495–525, 575–625 and 415–492 nm emission filters, and an objective lens, Plan-Aprochromat 63 x/1.4 oil DIC M27. During the image acquisition time, slides were kept at 37 °C in a humidified chamber (INUBTF-WSKM-F1, Tokai Hit, Japan). ZEN 2 software (Zeiss, Jena, Germany) was used for image processing. Co-localization quantification between DOX-emitted fluorescence (red) and LysoTracker^®^ Green DND-26 was done by calculating Pearson’s coefficient of correlation (PCC) with Imaris x 64 8.0.0 software. PCC values ranging between +1 (perfect correlation) and −1 (perfect negative correlation), where 0 means no correlation, were obtained [49]. PCC was calculated in regions of interest (ROI), separately defined for each image by masking the entire dataset with the red channel (DOX fluorescence). The value of PCC for each experiment is presented as mean ± S.D. calculated from three separate images.

#### 3.8.4. Flow Cytometry Measurements of Intracellular and Nuclear Accumulation of DOX from cPCPs

HT1080 cells were seeded in 6-well plates (flat-bottom cell culture microplate, 3505, Corning, Luzern, Switzerland) at a density of 150,000 cells/well and were left to attach overnight. The next day, cells were treated with serum-free medium containing 0.3 μM of DOX or equimolar concentrations of cPCP_4/5_^2^, cPCP_5_, cPCP_4/10_^2^ and cPCP_2/20_ (DOX eq). Cells were incubated for 24, 48 and 72 h in the dark at 37 °C. Cells were washed 3 times with DPBS, detached from the surface with a cell-scraper and resuspended in 500 μL of DPBS. Cells were analyzed immediately on a BD LSR Fortessa cell analyzer (BD Biosciences, San Jose, CA, USA). Excitation wavelength for DOX was set to 488 nm and emitted fluorescence was collected through a band pass filter between 564 and 606 nm.

For DOX uptake into the nucleus and cell cycle analysis, an adapted version of Vindolov’s reagent [50] was used to lyse the cells keeping the nuclei intact. For this purpose, 1 mM Tris buffer was prepared and the pH was adapted to 7.6 with 30% HCl. Igepal CA-630 and Hoechst 3342 (5 μg/mL final concentration) were added, and the reagent was filtered.

After the analysis of DOX cellular uptake, cells were centrifuged for 10 min at 1200 rpm and the pellet was resuspended in Vindolov’s reagent. Samples were incubated at 4 °C for 1 h and analyzed with a BDLSR Fortessa cell analyzer.

Acquisition data were analyzed using FlowJo V10 software (FLOWJO, LCC, Ashland, OR, USA). For cell cycle analysis, the Dean–Jett–Fox model was used. All measurements were done in triplicates. Mean values and S.D. were calculated and plotted for each tested compound. Fluorescence of control cells incubated in serum-free medium was subtracted from acquired DOX-emitted fluorescence.

All figures and statistical analysis were prepared using GraphPad Prism 7 software. One-way ANOVA was used to compare sample means. *p*-value < 0.05 was considered as statistically significant.

## 4. Conclusions

cPCPs composed from a cyclodecapeptide RAFT scaffold were successfully synthesized with the appropriate number of DOX—attached via a Cat B-cleavable peptide linker—and PEG moieties. Fluorescence quenching results showed a decrease in DOX fluorescence with an increased number of DOX molecules attached to the cPCP and molecular weight of PEG. Taken together, cell proliferation assays, confocal microscopy results and flow cytometry on HT1080 cells show that cPCPs exhibit a delayed action compared to free DOX that consists of a cell cycle arrest in the G2 phase and increased membrane stress. Moreover, our observations indicate that PEG 5, 10 and 20 kDa interfere with the release of DOX by Cat B in in vitro conditions, probably due to steric hindrance. Since the in vitro results do not necessarily correlate with the in vivo prodrug action, the potential of cPCPs should be further evaluated by an in vivo animal study.

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
