# Peer review of "Cathepsin B-Cleavable Cyclopeptidic Chemotherapeutic Prodrugs"

_molecules, 2020, doi:10.3390/molecules25184285_

Round 1
Reviewer 1 Report
Title: Cathepsin B-cleavable Cyclopeptidic Chemotherapeutic Prodrugs
Authors: Viktorija Herceg, Jordan Bouilloux, Karolina Janikowska, Eric Allémann and Norbert Lange
The authors reported the synthesis of doxorubicin-containing cyclopeptidic chemotherapeutic prodrugs with different doxorubicin-loading and PEGs. Total 4 conjugates were synthesized. Furthermore, a series of in vitro data for the conjugates were presented. This study fits the category of drug delivery. It is necessary to provide in vivo data, e.g. PK, TOX, tissue distribution, to show how the technique (RAFT) can improve the drawbacks of the original drug. However, only in vitro data are presented in the manuscript. If no in vivo data can be provided, the authors, at least, should predict the possible improvement of the in vivo properties from the in vitro data.
Author Response
Absence of in vivo data
The authors agree with reviewer 1’s comments. However, it was not possible due to the Swiss authorities to carry out these experiments. However, as proposed we have discussed already in the original manuscript in the last paragraph before the conclusions potential benefits of our new conjugates. If needed we can still add more discussions, but fear that we are going more into speculations than in founded hypotheses.
Reviewer 2 Report
See attached file.

Reviewer 3 Report
The manuscript by Herceg et al. regards the synthesis, fluorescence properties, cleavage and in vitro evaluation of Cathepsin B-cleavable cyclopeptidic Chemotherapeutic prodrugs.
The paper is well written, the topic is interesting and of great relevance in pharmaceutical field, and the choice of the journal is appropriate, so it surely deserves publication.
The English form quite good.
In my opinion, only the following minor revisions are required before the paper can be accepted for publication.
1) Paragraph 2.3: being the description of this part very detailed, the synthesis of DBCO-GAGRRAAG (Scheme S1) should be better reported in the main paper instead of the supplementary material. You can leave Figures S1, S2 and S3 in the supplementary material.
2) Paragraph 2.4: the same comment of point 1) applies to paragraph 2.4 “General procedure for the synthesis of the PEGylated cyclopeptidic conjugates”, Scheme S2 should also be reported in the main paper. You can leave the figures relative to this part in the supplementary material. Otherwise, being the description of this part very detailed, the comprehension is very difficult.
3) Paragraph 2.5: the same comments of points 1) and 2) applies also to paragraph 2.5 “General procedure for the synthesis of the pentasubstituted cyclopeptidic conjugate (see Scheme S3)”. Scheme S3 should be reported in the main paper, and you can leave the corresponding figures in the supplementary material.
4) Paragraph 3.1: please, put Scheme 1 before Table 1.
5) Paragraph 3.5: you should put Figure 5 after the line where it is cited in the text.
6) Paragraph 4: this part is too generic and not strictly related to a critical discussion of the reported results. It should be more adapt to an Introduction section, instead of a “Discussion” section. In the “Discussion” section the authors should report a critical interpretation of their data that, up to now, even if correct and well presented, appear as a simple list. They should avoid any general, well-known information, and comparison with literature should be made only in order to put into light how their study represents a step forward with respect to the state of art. Hence, the authors should rewrite this paragraph taking into account the aforementioned considerations. If they prefer, they could also eliminate this paragraph, reformulating the article structure and writing a single “Results and Discussion” paragraph, in which the critical interpretation of their data is done step by step.
Author Response
1. The scheme has been moved to Section 2.3. as suggested
All comments have been changed according to the reviewer’s suggestions
Round 2
Reviewer 1 Report
The authors have provided reasoable backgrouds and discussions for the possible in vivo potential of the compound. The manuscript can be accepted.